# Clinical Significance of Intratumoral Contrast Pooling on Contrast-Enhanced CT After Atezolizumab Plus Bevacizumab for Unresectable Hepatocellular Carcinoma

**DOI:** 10.3390/curroncol32120694

**Published:** 2025-12-09

**Authors:** Kiyoyuki Minamiguchi, Mariko Irizato, Ryota Nakano, Hideki Kunichika, Tetsuya Tachiiri, Ryosuke Taiji, Yuki Tsuji, Satoshi Yasuda, Hitoshi Yoshiji, Masayuki Sho, Toshihiro Tanaka

**Affiliations:** 1Department of Diagnostic and Interventional Radiology, Nara Medical University, 840 Shijyocho, Kashihara 634-8521, Nara, Japantotanaka@naramed-u.ac.jp (T.T.); 2Department of Gastroenterology, Nara Medical University, 840 Shijyocho, Kashihara 634-8521, Nara, Japan; 3Department of Surgery, Nara Medical University, 840 Shijyocho, Kashihara 634-8521, Nara, Japan; hi22zd@naramed-u.ac.jp (S.Y.);

**Keywords:** unresectable hepatocellular carcinoma, atezolizumab and bevacizumab, CT, immunotherapy

## Abstract

Recent advances in immunotherapy have improved outcomes for patients with unresectable hepatocellular carcinoma. Atezolizumab combined with bevacizumab has become a standard first-line regimen, but early imaging indicators of therapeutic efficacy remain unclear. We report two patients with unresectable hepatocellular carcinoma who showed patchy pooling of contrast material within the tumor on early follow-up CT after initiating atezolizumab plus bevacizumab therapy. This imaging feature, referred to as the “vascular lake-like phenomenon,” was subsequently followed by marked tumor regression and favorable clinical responses. The vascular lake-like phenomenon may therefore represent an early indicator of therapeutic efficacy with this combination therapy. Recognition of this distinctive imaging finding could help clinicians evaluate treatment response earlier.

## 1. Introduction

Advancements in immunotherapy have reshaped the treatment landscape for unresectable hepatocellular carcinoma (uHCC). Atezolizumab combined with bevacizumab (AB therapy) is recommended as a first-line treatment in the Barcelona Clinic Liver Cancer (BCLC) 2022 guidelines [1]. In parallel, other first-line options include durvalumab-tremelimumab (STRIDE) and ipilimumab-nivolumab, while multikinase inhibitors such as lenvatinib or sorafenib remain alternatives for patients ineligible for immunotherapy [2,3]. For those with progression on first-line regimens, additional multikinase inhibitors are available as subsequent options. Therefore, an unmet clinical need is to evaluate treatment efficacy as early as possible to support decisions to continue the current regimen or switch to an alternative therapy.

Adverse events associated with systemic therapy remain a major concern in clinical practice. A previous report described the pooling of contrast material within the tumor after the administration of AB therapy [4]. This finding has been interpreted as intratumoral hemorrhage and referred to as the vascular lake-like phenomenon. To our knowledge, however, no previous reports have focused on the clinical implications of this distinctive imaging feature following AB therapy. In this article, we report two cases of uHCC in which the pooling of contrast material was observed at the first response assessment following AB therapy, and was subsequently accompanied by tumor regression, suggesting that this finding may represent an early imaging biomarker of treatment efficacy with this combination therapy.

## 2. Case Presentation

### 2.1. Case 1

A woman in her seventies was referred to our hospital for evaluation of a hepatic mass, incidentally detected by ultrasonography during a workup for abnormal liver enzymes. She had a history of daily alcohol consumption (approximately 700 mL per day for 30 years). Laboratory data and tumor markers were as follows: aspartate aminotransferase (AST): 101 U/L, alanine aminotransferase (ALT): 627 U/L, lactate dehydrogenase (LDH): 235 U/L, alkaline phosphatase (ALP): 164 U/L, gamma-glutamyl transpeptidase (γ-GTP): 154 U/L, total bilirubin: 0.9 mg/dL, α-fetoprotein (AFP): 85.5 ng/mL, protein induced by vitamin K absence or antagonist-II (PIVKA-II): 30,056 mAU/mL, and carcinoembryonic antigen (CEA): 7.0 ng/mL. Serological testing for hepatitis B virus revealed negative results for HBV DNA. Anti-HBs was 14.7 mIU/mL, whereas anti-HBc was 0.35 S/CO. Contrast enhanced CT (CE-CT) demonstrated an approximately 10.3 cm tumor in the anterior segment of the liver, showing arterial-phase enhancement with a pseudocapsule, followed by washout on the portal and delayed phases (Figure 1).

This patient was diagnosed with uHCC. The Child–Pugh score was 5. AB therapy was selected as the first-line treatment for this patient. After four cycles of this combination therapy, tumor size decreased from 10.3 cm to 9.4 cm, and treatment response was categorized as stable disease (SD) according to Response Evaluation Criteria in Solid Tumors (RECIST) 1.1. At this time, patchy pooling of contrast medium within the tumor was observed. After seven cycles of this therapy, the tumor size was reduced to 4.3 cm, and the response was categorized as a partial response (PR) based on RECIST 1.1. After eleven cycles of this therapy, the patient underwent cTACE followed by RFA with curative intent, after which AB therapy resumed and continued to a total of 22 cycles.

### 2.2. Case 2

A woman in her eighties was referred to our hospital for evaluation of a hepatic mass, incidentally detected on ultrasonography during a workup for abnormal liver enzymes. Her medical history included hypertension and dyslipidemia. Laboratory data and tumor markers were as follows: AST: 54 U/L, ALT: 50 U/L, LDH: 185 U/L, ALP: 138 U/L, γ-GTP: 48 U/L, total bilirubin: 1.2 mg/dL, AFP: <0.9 ng/mL, PIVKA-II: 9795 mAU/mL, CEA: 4.2 ng/mL. Anti-HBs was 18.3 mIU/mL, whereas anti-HBc was 0.1 S/CO. The Child–Pugh score was 6. CE-CT revealed an approximately 14.5 cm tumor in segment 8 of the liver, showing arterial-phase enhancement with central necrosis, a pseudocapsule, and an intratumoral artery, followed by washout in the portal and delayed phases (Figure 2).

The tumor extended into the middle hepatic vein, forming a venous thrombus. AB therapy was initiated as the first-line treatment for this patient. After five cycles of this combination therapy, tumor size decreased from 14.5 cm to 7.0 cm, and treatment response was categorized as a PR according to RECIST 1.1. At this time, patchy pooling of contrast medium within the tumor was also observed. After seven cycles of this therapy, the response was categorized as a PR based on RECIST 1.1, and conversion therapy was planned.

## 3. Discussion

We report two cases of patients with uHCC who underwent AB therapy, in which follow-up CE-CT demonstrated persistent and localized pooling of contrast material within the tumor, followed by subsequent tumor regression during the clinical course. To our knowledge, no previous reports have focused on this distinctive imaging feature as a potential indicator of a favorable outcome with this combination therapy.

Only one prior report has described the same imaging finding in a patient with uHCC treated with AB therapy, namely the pooling of contrast material observed in a rib metastasis, which the authors termed the “vascular lake-like phenomenon” [4]. This phenomenon was initially proposed by Uchida et al. in the context of the multityrosine kinase inhibitor (lenvatinib) for HCC [5]. The vascular lake-like phenomenon resembles the “vascular lake phenomenon (VLP)” occasionally observed on angiography during TACE for HCC [6]. In TACE, a mechanical theory of VLP has been proposed, suggesting that rapid obstruction of tumor-feeding arteries increases the intratumoral pressure of the tumor microvasculature and leads to rupture and hemorrhage within the tumor. By contrast, with multityrosine kinase inhibitors, inhibition of vascular endothelial growth factor (VEGF) signaling suppresses tumor angiogenesis, promotes pseudoaneurysm formation with microrupture and subsequent intratumoral hemorrhage, reduces intratumoral blood flow, and causes tumor ischemia and necrosis [7]. Both lenvatinib and bevacizumab, which exert anti-VEGF activity, have been reported to be associated with intratumoral hemorrhage and the occurrence of the vascular lake-like phenomenon.

Previous studies of imaging biomarkers for AB therapy in uHCC have focused on pretreatment prediction based on baseline clinical and imaging characteristics [8,9,10]. These reports have shown that several baseline features are associated with early progression or treatment response to AB therapy, supporting the potential value of pretreatment imaging biomarkers for risk stratification. Given this emphasis, an important next step is to consider which HCCs are more likely to exhibit the vascular lake-like phenomenon. In our two cases, both patients had large HCCs with a pseudocapsule. Notably, large HCCs have a greater tendency to have intratumoral hemorrhage after lenvatinib administration, and several cases of HCCs larger than 9 cm have also been reported to exhibit the vascular lake-like phenomenon [5]. Although the underlying mechanism remains unknown, large HCCs may be more likely to develop the vascular lake-like phenomenon following AB therapy. Further validation in a large cohort will be required.

Recent studies have suggested that pretreatment 18F-fludeoxyglucose positron emission tomography/computed tomography (FDG-PET/CT), which mainly reflects glucose metabolism and does not directly reflect immune activity, may help identify HCCs that are more likely to benefit from combination immunotherapy [11,12,13]. Interestingly, in a recent cohort of FDG-PET/CT-positive HCCs treated with AB therapy, all cases showed at least an initial response, although a subset of patients subsequently showed re-elevation of tumor markers [13]. Therefore, combining our imaging finding with pretreatment FDG-PET/CT parameters may help to more precisely identify, earlier in the treatment course, patients who are expected to benefit from combination immunotherapy. Moreover, other reports have described PET tracers targeting prostate-specific membrane antigen (PSMA) and fibroblast activation protein inhibitor (FAPI) as useful for treatment response assessment and prognostic prediction in patients with HCC receiving immunotherapy, suggesting that PSMA- and FAPI-targeted PET could serve as complementary imaging modalities when combined with our imaging finding [14,15].

In consideration of our two cases, the vascular lake-like phenomenon may represent a favorable imaging sign following AB therapy. To date, its prognostic relevance has not been clarified. From an immunological perspective, the release of tumor-associated antigens following cancer cell death is essential to initiate and amplify the cancer immunity cycle [16]. We hypothesize that the vascular lake-like phenomenon is associated with tumor necrosis due to pseudoaneurysm formation and intratumoral hemorrhage and may lead to a greater release of tumor antigens into the systemic circulation (Figure 3). Such increased antigen exposure could further stimulate the cancer immune cycle and enhance the efficacy of immune checkpoint inhibition.

## 4. Conclusions

In conclusion, the appearance of the vascular lake-like phenomenon on early follow-up CE-CT after initiation of AB therapy may serve as an early indicator of therapeutic response in uHCC. Clinicians should recognize this imaging feature as a potential sign of favorable treatment efficacy. As this report describes only two cases, our findings should be interpreted with caution. Therefore, further prospective studies are warranted to validate its clinical significance.

## Figures and Tables

**Figure 1 curroncol-32-00694-f001:**
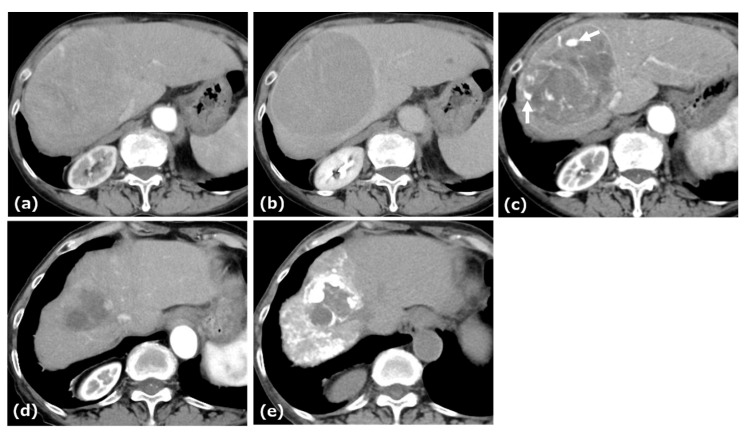
CE-CT of Case 1. CE-CT showed a large HCC in the anterior segment of the liver with arterial-phase hyperenhancement (**a**) and portal-phase washout with a surrounding pseudocapsule (**b**). After four cycles of AB therapy, arterial-phase CT demonstrated a vascular lake-like phenomenon ((**c**), arrow). On arterial-phase CT obtained after a total of seven cycles of AB therapy, tumor regression was observed (**d**). Locoregional therapy (conventional transarterial chemoembolization (cTACE) followed by radiofrequency ablation (RFA)) was subsequently performed with curative intent (**e**).

**Figure 2 curroncol-32-00694-f002:**
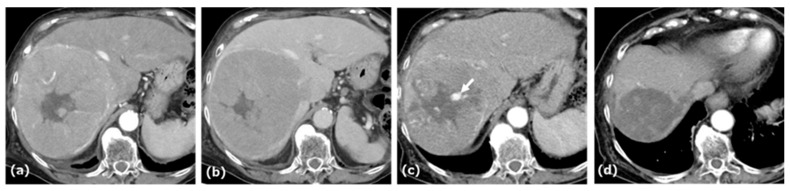
CE-CT of Case 2. CE-CT in the arterial and portal phases showed a large hypervascular HCC with central necrosis and a pseudocapsule in the anterior segment of the liver (**a**,**b**). After five cycles of AB therapy, arterial-phase CT demonstrated a vascular lake-like phenomenon ((**c**), arrow). On arterial-phase CT obtained after a total of seven cycles of AB therapy, tumor regression was observed (**d**).

**Figure 3 curroncol-32-00694-f003:**
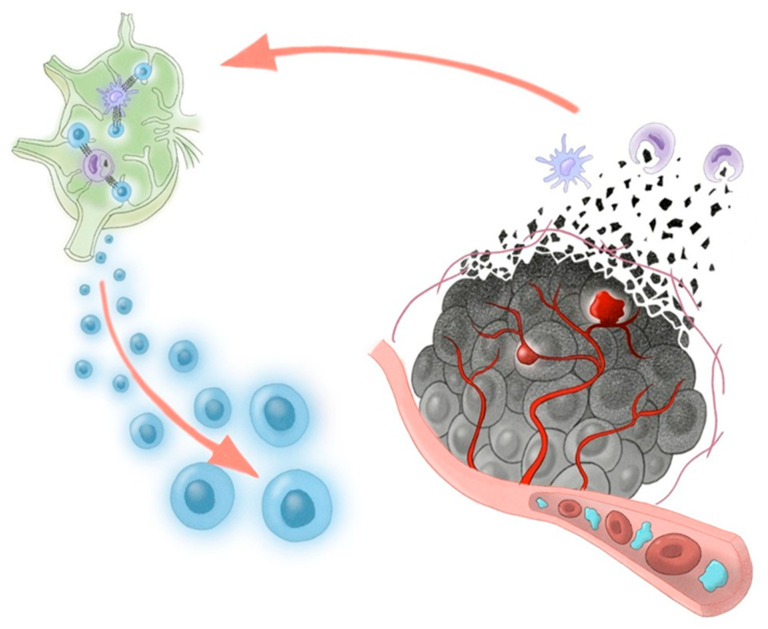
Proposed schema of the cancer-immunity cycle potentially triggered by AB therapy. Bevacizumab-induced pseudoaneurysm and intratumoral hemorrhage may increase the release of tumor-associated antigens, enabling lymph node T-cell priming. Activated T cells exert antitumor cytotoxicity within the tumor, and this effect is enhanced by PD-L1 blockade with atezolizumab.

## Data Availability

The datasets generated and/or analyzed during the current study are available from the corresponding author on reasonable request.

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
