# Peer review of "Clinical Significance of Intratumoral Contrast Pooling on Contrast-Enhanced CT After Atezolizumab Plus Bevacizumab for Unresectable Hepatocellular Carcinoma"

_curroncol, 2025, doi:10.3390/curroncol32120694_

Round 1

Reviewer 1 Report

Comments and Suggestions for Authors

The authors describe two informative case reports documenting patchy intratumoral contrast pooling (a “vascular lake-like phenomenon”) on early follow-up CT after atezolizumab + bevacizumab (AB) therapy, followed by marked tumor regression. The topic is clinically interesting because AB is widely used and early imaging indicators of response remain unclear. The manuscript is generally well written and the hypothesis is plausible. Given the small number of cases, claims should be appropriately tempered; nonetheless this is a valuable contribution as an early observation that merits publication after modest revision.

Some considerations:

  • Clarify Figure 2 (and Figure 1 as needed): panels c and d. The current Figure 2 caption and the manuscript text do not clearly explain what panels c and d represent (which contrast phase, exact timing relative to therapy, and whether images are from the same study/timepoint or different timepoints).
  • Add a short paragraph that explains why PSMA and FAPI PET could be useful in this clinical context (e.g., higher lesion detection sensitivity, ability to evaluate tumor biology and microenvironment, potential to detect residual viable tumor vs treatment-related changes), and explicitly state that PET could complement CT (not replace it) for response assessment or selection for conversion therapy.

Author Response

Dear Reviewer #1:

Thank you very much for your thoughtful review of our manuscript. Our point-by-point responses to your comments are provided below. We have also revised the manuscript accordingly based on your suggestions.

Specific comments:

Comment 1

Clarify Figure 2 (and Figure 1 as needed): panels c and d. The current Figure 2 caption and the manuscript text do not clearly explain what panels c and d represent (which contrast phase, exact timing relative to therapy, and whether images are from the same study/timepoint or different timepoints).

Reply 1

Thank you for your comment.

We have revised the Figure 1 and 2 legend as follows:

“Figure 1. CE-CT of Case 1. CE-CT showed a large HCC in the anterior segment of the liver with arterial-phase hyperenhancement (a) and portal-phase washout with a surrounding pseudocapsule (b). After four cycles of AB therapy, arterial-phase CT demonstrated a vascular lake-like phenomenon (c, arrow). On arterial-phase CT obtained after a total of seven cycles of AB therapy, tumor regression was observed (d). Locoregional therapy (conventional transarterial chemoembolization (cTACE) followed by radiofrequency ablation (RFA)) was subsequently performed with curative intent (e)”

“Figure 2. CE-CT of Case 2. CE-CT in the arterial and portal phases showed a large hypervascular HCC with central necrosis and a pseudocapsule in the anterior segment of the liver (a, b). After five cycles of AB therapy, arterial-phase CT demonstrated a vascular lake-like phenomenon (c, arrow). On arterial-phase CT obtained after a total of seven cycles of AB therapy, tumor regression was observed (d).”

Comment 2

Add a short paragraph that explains why PSMA and FAPI PET could be useful in this clinical context (e.g., higher lesion detection sensitivity, ability to evaluate tumor biology and microenvironment, potential to detect residual viable tumor vs treatment-related changes), and explicitly state that PET could complement CT (not replace it) for response assessment or selection for conversion therapy.

Reply 2

Thank you for your thoughtful comment.

We have added the following sentences in Discussion;

“Recent studies have suggested that pretreatment 18F-fludeoxyglucose positron emission tomography/computed tomography (FDG-PET/CT), which mainly reflects glucose metabolism and does not directly reflect immune activity, may help identify HCCs that are more likely to benefit from combination immunotherapy (1-3). Interestingly, in a recent cohort of FDG-PET/CT-positive HCCs treated with AB therapy, all cases showed at least an initial response, although a subset of patients subsequently showed re-elevation of tumor markers (3). Therefore, combining our imaging finding with pretreatment FDG-PET/CT parameters may help to more precisely identify, earlier in the treatment course, patients who are expected to benefit from combination immunotherapy. Moreover, other reports have described PET tracers targeting prostate-specific membrane antigen (PSMA) and fibroblast activation protein inhibitor (FAPI) as useful for treatment response assessment and prognostic prediction in patients with HCC receiving immunotherapy, suggesting that PSMA- and FAPI- targeted PET could serve as complementary imaging modalities when combined with our imaging finding (4-5).”

(1) Wang G, Zhang W, Chen J, Luan X, Wang Z, Wang Y, Xu X, Yao S, Guan Z, Tian J, et al. Pretreatment Metabolic Parameters Measured by 18F-FDG PET to Predict the Pathological Treatment Response of HCC Patients Treated With PD-1 Inhibitors and Lenvatinib as a Conversion Therapy in BCLC Stage C. Front Oncol. 2022;12:884372. doi: 10.3389/fonc.2022.884372. PMID: 35719917; PMCID: PMC9204225.

(2) Lee JW, Lee SM, Kang B, Kim JS, An C, Chon HJ, Jang SJ. Prognostic Significance of Volumetric Parameters on Pretreatment FDG PET/CT in Patients With Hepatocellular Carcinoma Receiving Atezolizumab Plus Bevacizumab Therapy. Clin Nucl Med. 2025;50:486-494. doi:10.1097/RLU.0000000000005896. Epub 2025 Apr 21. PMID: 40254801.

(3) Kudo M. Fluorine-18 Fluorodeoxyglucose Positron Emission Tomography: A Potential Imaging Biomarker for Predicting Response to Combination Immunotherapy in Hepatocellular Carcinoma. Liver Cancer. 2025;14:511-517.

(4) Antony A, Tran NH, Patnam NG, Trivedi KH, Karbhari A, Garima S, Mukherjee S, Jacobson MS, Kemp BJ, Thompson SM et al. Interpretive consistency of a qualitative 68Ga-PSMA PET framework for treatment response assessment in hepatocellular carcinoma: a head-to-head comparison with cross-sectional imaging. Eur J Nucl Med Mol Imaging. 2025 Nov 6. doi: 10.1007/s00259-025-07629-w. Epub ahead of print. PMID: 41193901.

(5) Wu M, Wang Y, Yang Q, Wang X, Yang X, Xing H, Sang X, Li X, Zhao H, Huo L. Comparison of Baseline 68Ga-FAPI and 18F-FDG PET/CT for Prediction of Response and Clinical Outcome in Patients with Unresectable Hepatocellular Carcinoma Treated with PD-1 Inhibitor and Lenvatinib. J Nucl Med. 2023;64:1532-1539. doi: 10.2967/jnumed.123.265712. Epub 2023 Jul 27. PMID: 37500263.

Reviewer 2 Report

Comments and Suggestions for Authors

Dear Editor-In-Chief

Current Oncology, MDPI,

Subject: Review of the article curroncol-4000955

Entitled “Clinical Significance of Intratumoral Contrast Pooling on Con-trast-Enhanced CT After Atezolizumab Plus Bevacizumab for Unresectable Hepatocellular Carcinoma”

Early imaging biomarkers of treatment efficacy are critical to prevent delays in determining whether to continue the current regimen or transition to an alternative therapy for patients with unresectable hepatocellular carcinoma (uHCC). The present study reports two uHCC cases that exhibited patchy pooling of contrast material within the tumor on early follow-up contrast-enhanced CT after initiation of AB therapy. The work is both relevant and important; however, its reliability remains uncertain. The study could be further strengthened on several levels. Below, I have outlined general and specific comments for the authors’ consideration

General comments

  1. In case #1, a stable response was noted after 4 cycles of AB therapy, whereas in case #2, a partial response was observed after 5 cycles of AB therapy. Given the variability in individual responses, were any additional biomarkers evaluated to help predict response type? In other words, was a correlation study performed? I understand the limitation of having only two cases.
  2. Considering the tumor progression observed in the two cases treated with AB therapy, how does this outcome compare with other first-line treatment options?
  3. The study reports a partial response in both cases, yet also presents evidence of a complete response. This inconsistency is confusing to the reviewer.

Specific comments:

  1. the title: it is contrast not con-trast.
  2. keywords: suggesting to add immunotherapy; unresectable hepatocellular carcinoma.
  3. figure 1: change the layout to ese the visual comparison, move b to the top and e to the bottom. However, if d is supposed to demonstrate the complete response, then the order of the images is incorrect! No description for image e in the figure caption?
  4. line 84: cTACE and RFA are acronyms that need to be spelled out at first appearance. Likewise, RECIST in line 89.
  5. line 91, the text refers to a partial response—could you clarify the extent of tumor size reduction? Additionally, is the complete response illustrated in image D of Figure 1, and was RECIST 1.1 applied to categorize the response?
  6. fig 2: image d demonstrates tumor regression? No description for image d in the figure caption.
  7. lines 139-140: Including a summary of studies 8 to 10 will help connect the readers’ knowledge flow and strengthen the discussion.
  8. lines 152-154: would performing a PET/CT scan help to confirm the authors’ hypothesis?
  9. any potential limitation that the authors should declare?
  10. very low number of references ..

Author Response

Dear Reviewer #2:

Thank you very much for your thoughtful review of our manuscript. Our point-by-point responses to your comments are provided below. We have also revised the manuscript accordingly based on your suggestions.

Specific comments:

Comment 1

General comments

In case #1, a stable response was noted after 4 cycles of AB therapy, whereas in case #2, a partial response was observed after 5 cycles of AB therapy. Given the variability in individual responses, were any additional biomarkers evaluated to help predict response type? In other words, was a correlation study performed? I understand the limitation of having only two cases.

Reply 1

Thank you for your thoughtful comment.

As you mentioned, our manuscript describes only two cases, and it is therefore difficult to examine and determine reliable predictors of response. Based on the clinical, laboratory, and imaging data available for these patients, we did not identify any obvious features that could clearly suggest the response type in these two cases.

Comment 2

Considering the tumor progression observed in the two cases treated with AB therapy, how does this outcome compare with other first-line treatment options?

Reply 2

Thank you for your important comment.

Our aim is to describe the characteristic imaging finding of a vascular lake-like phenomenon and to discuss its possible association with subsequent tumor response, rather than to compare outcomes between different first-line treatment options. As shown in Figure 3, we hypothesize that this phenomenon is initiated by VEGF inhibition, and that subsequent PD-L1 blockade enhances antitumor immune cytotoxicity. This imaging finding may not be expected with first-line regimens that do not include an anti-VEGF inhibitor, such as combinations of immune checkpoint inhibitors alone.

Comment 3

The study reports a partial response in both cases, yet also presents evidence of a complete response. This inconsistency is confusing to the reviewer.

Reply 3

Thank you for the valuable comment.

To avoid confusion between partial response to AB therapy and the subsequent clinical course, we deleted the sentence “A complete response has been maintained thereafter” from the description of Case 1 and revised it to: “After eleven cycles of this therapy, the patient underwent cTACE followed by RFA with curative intent, after which AB therapy was resumed and continued to a total of 22 cycles.”

We have also revised the Figure 1 legend by replacing the phrase “with the aim of achieving a complete response” with “with curative intent.”

Comment 4

Specific comments:

the title: it is contrast not con-trast.

Reply 4

We have changed from “con-trast” to “contrast” in Title.

Comment 5

keywords: suggesting to add immunotherapy; unresectable hepatocellular carcinoma.

Reply 5

We have added “immunotherapy” and “unresectable hepatocellular carcinoma” in Keywords.

Comment 6

figure 1: change the layout to ese the visual comparison, move b to the top and e to the bottom. However, if d is supposed to demonstrate the complete response, then the order of the images is incorrect! No description for image e in the figure caption?

Reply 6

We have modified the layout of Figure 1 so that (b) has been moved to the top row and (e) to the bottom row. In addition, we have revised the figure including (e) as follows:

“Locoregional therapy (conventional transarterial chemoembolization (cTACE) followed by radiofrequency abla-tion (RFA)) was subsequently performed with curative intent (e).”

Comment 7

line 84: cTACE and RFA are acronyms that need to be spelled out at first appearance. Likewise, RECIST in line 89.

Reply 7

We have spelled out these abbreviations at their first appearance in the manuscript as follows: “conventional transarterial chemoembolization (cTACE)”, “radiofrequency ablation (RFA)”, and “Response Evaluation Criteria in Solid Tumors (RECIST)”.

Comment 8

line 91, the text refers to a partial response—could you clarify the extent of tumor size reduction? Additionally, is the complete response illustrated in image D of Figure 1, and was RECIST 1.1 applied to categorize the response?

Reply 8

Thank you for your comment.

(d) in Figure 1 represents a partial response. To avoid misunderstanding, we have revised the Figure 1 legend as follows:

“Figure 1. CE-CT of Case 1. CE-CT showed a large HCC in the anterior segment of the liver with ar-terial-phase hyperenhancement (a) and portal-phase washout with a surrounding pseudocapsule (b). After four cycles of AB therapy, arterial-phase CT demonstrated a vascular lake-like phe-nomenon (c, arrow). On arterial-phase CT obtained after a total of seven cycles of AB therapy, tumor regression was observed (d). Locoregional therapy (conventional transarterial chemoem-bolization (cTACE) followed by radiofrequency ablation (RFA)) was subsequently performed with curative intent (e).”

We have also revised the original sentence to clarify the extent of tumor size reduction in the description of Case 1 as follows:

“After seven cycles of this therapy, the tumor size was reduced to 4.3 cm, and the response was categorized as a partial response (PR) based on RECIST 1.1.”

Comment 9

fig 2: image d demonstrates tumor regression? No description for image d in the figure caption.

Reply 9

Thank you for your comment.

We have revised the Figure 2 legend as follows:

“Figure 2. CE-CT of Case 2. CE-CT in the arterial and portal phases showed a large hypervascular HCC with central necrosis and a pseudocapsule in the anterior segment of the liver (a, b). After five cycles of AB therapy, arterial-phase CT demonstrated a vascular lake-like phenomenon (c, arrow). On arterial-phase CT obtained after a total of seven cycles of AB therapy, tumor regression was observed (d).”

Comment 10

lines 139-140: Including a summary of studies 8 to 10 will help connect the readers’ knowledge flow and strengthen the discussion.

Reply10

To improve the knowledge flow and strengthen the Discussion, we have added a brief summary of these studies in the Discussion as follows:

“Previous studies of imaging biomarkers for AB therapy in uHCC have focused on pretreatment prediction based on baseline clinical and imaging characteristics [8-10]. These reports have shown that several baseline features are associated with early progression or treatment response to AB therapy, supporting the potential value of pretreatment imaging biomarkers for risk stratification.”

Comment 11

lines 152-154: would performing a PET/CT scan help to confirm the authors’ hypothesis?

Reply 11

Thank you for your comment.

FDG-PET mainly reflects glucose metabolism and does not directly reflect immune activity. Therefore, FDG-PET would not be expected to directly confirm our current hypothesis. However, several recent studies have reported pretreatment FDG-PET as a useful imaging biomarker for predicting response and prognosis in patients with HCC receiving combination immunotherapy. We have added the following sentences in Discussion;

“Recent studies have suggested that pretreatment 18F-fludeoxyglucose positron emission tomography/computed tomography (FDG-PET/CT), which mainly reflects glucose metabolism and does not directly reflect immune activity, may help identify HCCs that are more likely to benefit from combination immunotherapy (1-3). Interestingly, in a recent cohort of FDG-PET/CT-positive HCCs treated with AB therapy, all cases showed at least an initial response, although a subset of patients subsequently showed re-elevation of tumor markers (3). Therefore, combining our imaging finding with pretreatment FDG-PET/CT parameters may help to more precisely identify, earlier in the treatment course, patients who are expected to benefit from combination immunotherapy.”

(1) Wang G, Zhang W, Chen J, Luan X, Wang Z, Wang Y, Xu X, Yao S, Guan Z, Tian J, et al. Pretreatment Metabolic Parameters Measured by 18F-FDG PET to Predict the Pathological Treatment Response of HCC Patients Treated With PD-1 Inhibitors and Lenvatinib as a Conversion Therapy in BCLC Stage C. Front Oncol. 2022;12:884372. doi: 10.3389/fonc.2022.884372. PMID: 35719917; PMCID: PMC9204225.

(2) Lee JW, Lee SM, Kang B, Kim JS, An C, Chon HJ, Jang SJ. Prognostic Significance of Volumetric Parameters on Pretreatment FDG PET/CT in Patients With Hepatocellular Carcinoma Receiving Atezolizumab Plus Bevacizumab Therapy. Clin Nucl Med. 2025;50:486-494. doi:10.1097/RLU.0000000000005896. Epub 2025 Apr 21. PMID: 40254801.

(3) Kudo M. Fluorine-18 Fluorodeoxyglucose Positron Emission Tomography: A Potential Imaging Biomarker for Predicting Response to Combination Immunotherapy in Hepatocellular Carcinoma. Liver Cancer. 2025;14:511-517. doi:10.1159/000547990. PMID: 41122677; PMCID: PMC12536241.

Comment 12

any potential limitation that the authors should declare?

Reply 12

Thank you for this important comment.

We have revised the Conclusion section as follows:

“As this report describes only two cases, our findings should be interpreted with caution. Therefore, further prospective studies are warranted to validate its clinical significance.”

Comment 13

very low number of references ..

Reply 13

We have added the following references:

(1) Wang G, Zhang W, Chen J, Luan X, Wang Z, Wang Y, Xu X, Yao S, Guan Z, Tian J, et al. Pretreatment Metabolic Parameters Measured by 18F-FDG PET to Predict the Pathological Treatment Response of HCC Patients Treated With PD-1 Inhibitors and Lenvatinib as a Conversion Therapy in BCLC Stage C. Front Oncol. 2022;12:884372. doi: 10.3389/fonc.2022.884372. PMID: 35719917; PMCID: PMC9204225.

(2) Lee JW, Lee SM, Kang B, Kim JS, An C, Chon HJ, Jang SJ. Prognostic Significance of Volumetric Parameters on Pretreatment FDG PET/CT in Patients With Hepatocellular Carcinoma Receiving Atezolizumab Plus Bevacizumab Therapy. Clin Nucl Med. 2025;50:486-494. doi:10.1097/RLU.0000000000005896. Epub 2025 Apr 21. PMID: 40254801.

(3) Kudo M. Fluorine-18 Fluorodeoxyglucose Positron Emission Tomography: A Potential Imaging Biomarker for Predicting Response to Combination Immunotherapy in Hepatocellular Carcinoma. Liver Cancer. 2025;14:511-517. doi:10.1159/000547990. PMID: 41122677; PMCID: PMC12536241.

(4) Antony A, Tran NH, Patnam NG, Trivedi KH, Karbhari A, Garima S, Mukherjee S, Jacobson MS, Kemp BJ, Thompson SM et al. Interpretive consistency of a qualitative 68Ga-PSMA PET framework for treatment response assessment in hepatocellular carcinoma: a head-to-head comparison with cross-sectional imaging. Eur J Nucl Med Mol Imaging. 2025 Nov 6. doi: 10.1007/s00259-025-07629-w. Epub ahead of print. PMID: 41193901.

(5) Wu M, Wang Y, Yang Q, Wang X, Yang X, Xing H, Sang X, Li X, Zhao H, Huo L. Comparison of Baseline 68Ga-FAPI and 18F-FDG PET/CT for Prediction of Response and Clinical Outcome in Patients with Unresectable Hepatocellular Carcinoma Treated with PD-1 Inhibitor and Lenvatinib. J Nucl Med. 2023;64:1532-1539. doi: 10.2967/jnumed.123.265712. Epub 2023 Jul 27. PMID: 37500263.

Round 2

Reviewer 1 Report

Comments and Suggestions for Authors

The authors properly addressed my concerns.